

# Thinning leads to calving-style changes at Bowdoin Glacier, Greenland

Eef C. H. van Dongen[1], Guillaume Jouvet[2,3], Shin Sugiyama[4], Evgeny A. Podolskiy[4], Martin Funk[1], Douglas I. Benn[5], Fabian Lindner[1,6], Andreas Bauder[1], Julien Seguinot[1], Silvan Leinss[7], and Fabian Walter[1]

[1]Laboratory of Hydraulics, Hydrology and Glaciology (VAW), ETH Zurich, Zürich, Switzerland
[2]Department of Geography, University of Zurich, Zürich, Switzerland
[3]Autonomous Systems Laboratory, ETH Zurich, Zürich, Switzerland
[4]Institute of Low Temperature Science, Hokkaido University, Sapporo, Japan
[5]Department of Geography and Sustainable Development, University of St Andrews, St. Andrews, UK
[6]Department of Earth and Environmental Sciences, Ludwig-Maximilians-University, München, Germany
[7]Institute of Environmental Engineering, ETH Zurich, Zürich, Switzerland

**Correspondence:** Eef van Dongen (vandongen@vaw.baug.ethz.ch)

**Abstract.** Ice mass loss from the Greenland Ice Sheet is the largest single contributor to sea-level rise in the 21st century. The mass loss rate has accelerated in recent decades mainly due to thinning and retreat of its outlet glaciers. The diverse calving mechanisms responsible for tidewater glacier retreat are not fully understood yet. Since a tidewater glacier's sensitivity to external forcings depends on its calving style, a detailed insight into calving processes is necessary to improve projections of ice sheet mass loss by calving. As tidewater glaciers are mostly thinning, their calving styles are expected to change. Here,

5    we study calving behaviour changes under a thinning regime at Bowdoin Glacier, Northwest Greenland, by combining field and remote sensing data from 2015 to 2019. Previous studies showed that major calving events in 2015 and 2017 were driven by hydro-fracturing and melt-undercutting. New observations from UAV imagery and a GPS network installed at the calving front in 2019 suggest ungrounding and buoyant calving have recently occurred, as they show (1) increasing tidal modulation of vertical motion compared to previous years, (2) absence of a surface crevasse prior to calving, and (3) uplift and horizontal

10   surface compression prior to calving. Furthermore, an inventory of calving events from 2015 to 2019 based on satellite imagery provides additional support for a change towards buoyant calving since it shows an increasing occurrence of calving events outside of the melt season. The observed change of calving style could lead to a possible retreat of the terminus, which has been stable since 2013. We therefore highlight the need for high-resolution monitoring to detect changing calving styles and

15   numerical models that cover the full spectrum of calving mechanisms to improve projections of ice sheet mass loss by calving.

## 1 Introduction

Greenland's tidewater glaciers have been observed to accelerate, thin and retreat faster than any other part of the ice sheet (e.g. Pritchard et al., 2009; Hill et al., 2017; IMBIE Team, 2019). There is a large potential for rapid retreat in the northwest, where retreat and discharge have continued to accelerate within the past three years (King et al., 2020). projections of future ice sheet



mass loss, and thereby of sea level rise, depend on the capability to predict tidewater glacier behaviour. Despite recent advances in observing and modelling calving mechanisms, fundamental gaps in our understanding of outlet glacier sensitivity to climate change remain, such as quantification of ice–ocean interaction and understanding of glacier behaviour under near-buoyant and super-buoyant conditions (Benn and Åström, 2018; Catania et al., 2020).

Observations show that for some tidewater glaciers, large-scale infrequent calving events dominate the ice mass loss over small frequent events (Walter et al., 2010; James et al., 2014; Åström et al., 2014; Medrzycka et al., 2016). We use the term 'large-scale' relative to each individual glacier's annual mass loss through calving. As the physical processes triggering small- and large-scale events differ (Medrzycka et al., 2016), it is particularly important to understand the mechanisms behind large-scale events. Benn and Åström (2018) describe four main calving mechanisms:

1. **Rifting due to extensional stress:** stretching in response to large-scale velocity gradients initiates crevasses which penetrate to full depth (rifts). The rifts can propagate over days to weeks and eventually release large icebergs. On tidewater glaciers, fracture-inducing velocity gradients exist towards the terminus due to increasing basal motion (because effective pressure, and thus drag, is reduced at the bed as the glacier approaches flotation towards the terminus), and reduced lateral drag if ice flows into a wider part of a fjord. Propagation of crevasses to rifts requires the presence of water to oppose the stabilizing effect of ice overburden pressure (hydro-fracturing). If sufficient meltwater is available, a surface crevasse may penetrate through the glacier thickness (Benn et al., 2007). Basal water pressure can promote basal crevasse propagation if ice is at or close to flotation (Benn and Åström, 2018).

2. **Stresses associated with the force imbalance at ice fronts:** the outward-directed cryostatic pressure is greater than the backward-directed hydrostatic pressure. This imbalance induces a deviatoric stress which increases with ice cliff height (Hanson and Hooke, 2000). This stress leads to crevasse formation and may exceed the strength of ice if cliffs are tall enough (freeboard between 100 and 285 m, Parizek et al., 2019), a mechanism called the marine ice cliff instability (Pollard et al., 2015).

3. **Undercutting by subaqueous melting:** removal of support from the calving front. Undercut calving fronts have been observed on Greenlandic tidewater glaciers and near plumes, which release buoyant meltwater subglacially (Fried et al., 2015; Rignot et al., 2015). Depending on the ice-front geometry, iceberg sizes may exceed the extent of melt-induced undercuts, a so-called 'calving amplifier effect' (O'Leary and Christoffersen, 2013). Analysis of stresses suggests that calving rates may be up to four times the subaqueous melt rate if undercuts are sufficiently large (Benn et al., 2017). However, if frequent small calving events repeatedly remove the destabilized ice, such as observed on Tunabreen (Svalbard, How et al., 2019), the undercut may never become large enough to induce a calving amplifier effect.

4. **Buoyant calving:** uplift of a super-buoyant glacier tongue. Ice flow into deep water can result in a terminus below buoyant equilibrium ('super-buoyancy') and thus subjected to upward-directed buoyant forces. These forces may lead to upward rotation of the terminus, which can create flexure that induces basal crevasse formation and detachment of full thickness icebergs of hundreds of metres lateral extent (Murray et al., 2015). Numerical simulations by Benn et al.





(2017) indicate that super-buoyant conditions can also develop by thinning due to accelerated longitudinal stretching, known as dynamic thinning.

Which calving mechanisms dominate depends on the glacier geometry and environmental forcings (Benn and Åström, 2018). Observations show that calving events can involve several processes both in time and space for a single glacier. At Eqip Sermia (Greenland), inhomogeneous geometry causes different calving processes to occur; calving events are more frequent and larger where the fjord is shallow than for the deep part which has a smaller ice cliff above sea level (Walter et al., 2020). When Columbia Glacier (Alaska) became ungrounded in 2007, a calving-style transition occurred from a steady release of

low-volume icebergs, to episodic flow-perpendicular rifting, and release of very large icebergs (Walter et al., 2010).

Calving styles also vary seasonally, as most tidewater glaciers on Greenland advance in winter and retreat in summer (e.g. Moon and Joughin, 2008). Potential drivers of seasonal behaviour include ice mélange buttressing, increased surface melt leading to hydro-fracturing and acceleration of ice flow, and increased undercutting. Fried et al. (2018) studied the drivers of seasonal terminus change at 13 central west Greenland tidewater glaciers. They found that seasonal cycles are mainly governed

by variations in glacial runoff, rather than ice mélange or ocean thermal forcing, but the sensitivity to forcings depends on the dominant calving style. Sakakibara and Sugiyama (2020) studied seasonal cycles of ten tidewater glaciers in Northwest Greenland. Three large, fast-flowing glaciers had a nearly constant frontal ablation rate, except for several large calving events which occurred irrespective of ice-mélange conditions. On the contrary, around 50% of the annual frontal ablation occurred during summer at four slower flowing glaciers (including Bowdoin Glacier, the focus of this study). Sakakibara and Sugiyama (2020)

relate these differences to the fjord depth which is larger at the fast-flowing glaciers. Whereas warm Atlantic water intrudes into deep fjords and induces submarine melt throughout the year (Porter et al., 2014), the submarine melt rate increases in summer in shallow fjords because of fjord circulation driven by subglacial discharge (Sciascia et al., 2013). Thereby, increased submarine melt leads to larger frontal ablation in summer for the glaciers which terminate in shallower fjords.

With the sustained thinning as observed at the margins of the Greenland Ice Sheet (IMBIE Team, 2019), ungrounding of

termini is expected to affect calving processes. In order to improve projections of mass loss by calving, a better understanding of changing calving mechanisms is needed. To study the short-term variations in flow and flotation conditions which are essential to analyse calving mechanisms in detail, high temporal and spatial resolution ice motion data is required which cannot be provided by remote sensing (Podrasky et al., 2014). However, field observations near the calving fronts of tidewater glaciers are logistically challenging and therefore limited. An exception is the study by Murray et al. (2015), who captured glacier

dynamics at high temporal resolution, by deploying a GPS network at the calving front of Helheim Glacier. They recorded glacier motion during a number of major buoyant calving events.

Here, we study calving mechanisms under a thinning regime at Bowdoin Glacier, a 2.8 km wide tidewater glacier in Northwest Greenland (Fig. 1a). Its terminus was grounded but near flotation in 2013 (Sugiyama et al., 2015). We combine five years of field and remote sensing data to investigate whether a change of calving mechanisms may have occurred at Bowdoin

Glacier. We present measurements from a network of 10 GPS stations installed along the front in July 2019. The GPS network monitored glacier motion during a calving event on 20 July and prior to a second calving event on 29 July. First, we analyse the GPS data and show that these two calving events were likely triggered by buoyancy-driven basal crevassing. Then, we compose





an inventory of large-scale calving events from satellite imagery which shows that calving events seem to have become more frequent outside the melting season.

## 2    Characteristics of Bowdoin Glacier

Bowdoin Glacier (77°N, 68°W, Kangerluarsuup Sermia in Greenlandic, Bjørk et al., 2015) provides a unique opportunity to study calving through in situ measurements at the calving front, due to its accessibility via a crevasse-free walkable medial moraine (Fig. 1a). Field campaigns on Bowdoin Glacier were repeated in the summers of 2013–2017 and 2019 in a collaboration between Hokkaido University, ETH Zurich and University of Florence (Podolskiy et al., 2017; Seguinot et al., 2020).

Bowdoin Glacier dynamically thinned at a rate greater than $5 \mathrm{\,m\,yr^{-1}}$ in 2007–2013 (Tsutaki et al., 2016; Sakakibara and Sugiyama, 2018). From 2008, Bowdoin Glacier retreated rapidly after more than 20 years of a fairly stable terminus position on a bump on the ocean floor (Sugiyama et al., 2015). It stabilized to its current position in 2013, with a near-buoyant terminus, on inland-sloping bedrock upstream the bump (Fig. 1). The glacier was up to $250 \mathrm{\,m}$ thick at the calving front in 2013 (Sugiyama et al., 2015). Since 2013, the ice flow velocity did not decrease (Sakakibara and Sugiyama, 2020), therefore further thinning and ungrounding of the terminus are expected.

Flow velocities vary seasonally. In 2015 they ranged from 0.8 to $1.1 \mathrm{\,m\,d^{-1}}$ from March to May, increasing to $1.6 \mathrm{\,m\,d^{-1}}$ in early July and decreasing again to $1 \mathrm{\,m\,d^{-1}}$ in September (Sakakibara and Sugiyama, 2020). The marginal area southeast of the medial moraine is almost stagnant, creating a zone of high shear in the vicinity of the moraine. The slow moving area is characterized by shallow bedrock (Jouvet et al., 2017; van Dongen et al., 2020a). The high-shear zone acts as a 'suture zone', where crevasse tips arrest (Hulbe et al., 2010), such that the area in close vicinity to the medial moraine is usually crevasse-free.

Kilometre-scale calving events form a major part of Bowdoin Glacier's mass loss by calving (Jouvet et al., 2017; Minowa et al., 2019). During field campaigns in 2015 and 2017, deep surface crevasses were observed which lead to major calving events at similar locations near the moraine, where a plume was visible (Fig. , 2, 3a,b). A supraglacial meltwater stream has been present along the moraine during the field campaigns. In 2015 and 2017, the stream discharged into the surface crevasses as soon as they formed, deepening them by hydro-fracturing, leading to calving 5–15 days later. Numerical modelling showed that the crevasses were at least half-thickness deep, and that melt-undercutting in vicinity of the plume likely facilitated the 2017 calving (Jouvet et al., 2017; van Dongen et al., 2020a, b).

## 3    Data and methods

We study effects of glacier thinning on Bowdoin Glacier's calving behaviour. To analyse drivers of fracture initiation in detail, we installed a GPS network at the calving front in July 2019, which provides short-term variations in vertical motion and strain. A summary of events during the field campaign is provided in Table 1. Satellite and timelapse imagery are used to construct a calving event inventory to investigate possible changes in Bowdoin's calving mechanisms since 2015. Table 2 provides an overview of the periods for which each data set is available.
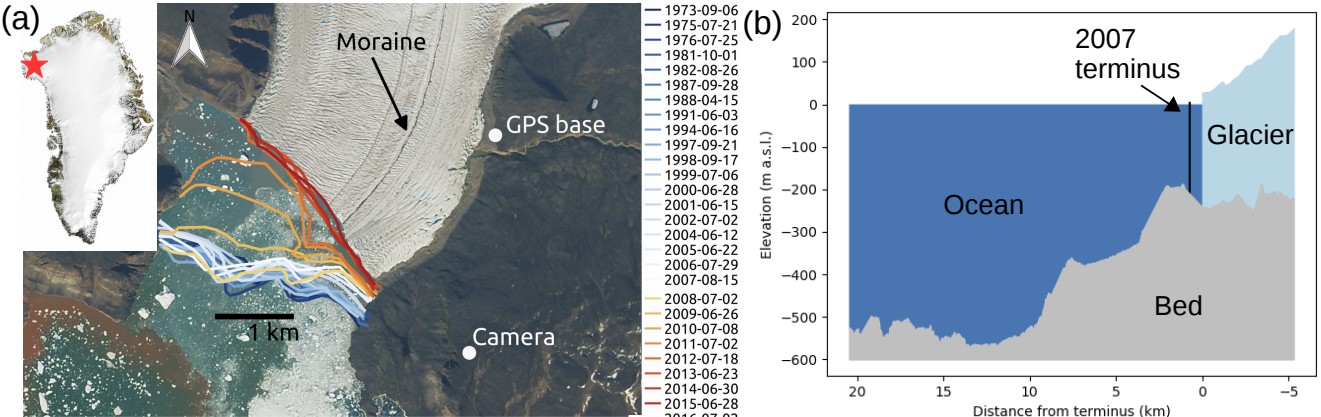

**Figure 1.** (a) Map of Bowdoin Glacier, the star in the upper left inset indicates the position of Bowdoin Glacier in Greenland (MODIS, ORNL DAAC, 2018) and the Copernicus Sentinel-2 satellite image shows Bowdoin Glacier on 1 July 2019. White dots show the positions of the timelapse camera and GPS base station. Front positions from 1973 to 2016 in varying colour (see legend), derived from satellite imagery, are also shown (Sugiyama et al., 2015; Sakakibara and Sugiyama, 2018). (b) Center-line cross section of ocean bed, glacier bed and glacier surface elevations as in summer 2013, including the front position in 2007 (Sugiyama et al., 2015).

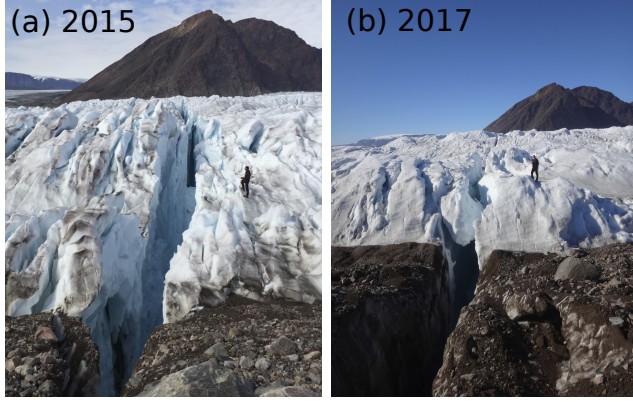

**Figure 2.** Pictures of the deep crevasses at the calving front in July 2015 (a, date unknown) and 6 July 2017 (b) that both led to large-scale calving events, both with a person for scale (a, credit: E. Podolskiy and b, credit: L. Preiswerk).

## 3.1 Elevation change

120   To study ungrounding, glacier thinning was derived from ArticDEMs of 2014–2017 (Porter et al., 2018). The digital elevation models (DEMs) were coregistered using the method outlined in Nuth and Kääb (2011) and implemented in the GeoUtils library (http://github.com/GeoUtils). Additionally, a 2D linear function is estimated from the elevation difference over stable areas and removed from the map of elevation change.



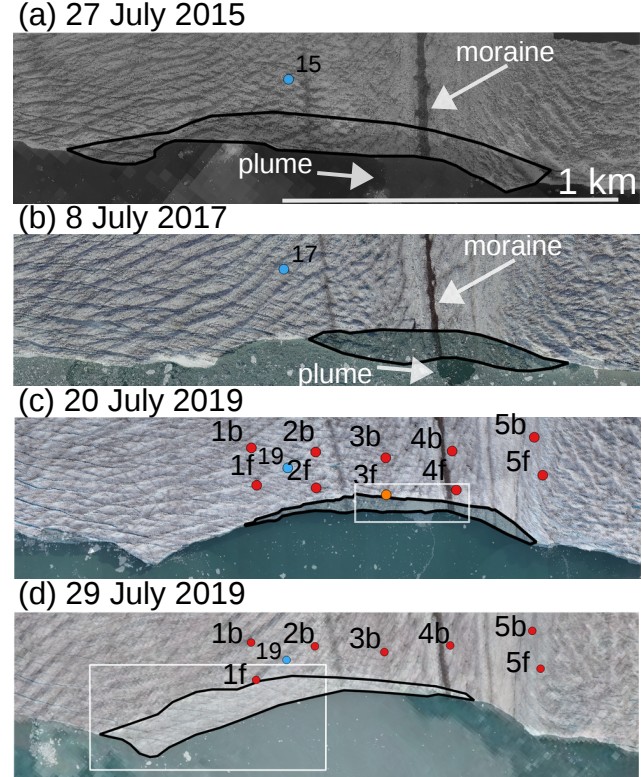

**Figure 3.** UAV-derived ortho-images of the calving front of Bowdoin Glacier showing large calving events that occurred on 27 July 2015 (a), 8 July 2017 (b), 20 July 2019 (c) and 29 July 2019 (d). All panels have the same scale and cover the same part of the front (outlined in Fig. 4a). Pre-calving images are overlayed by 50% transparent post-calving images. Red dots show the GPS stations installed in 2019 (c and d, with an orange dot for station 3f that calved off). Blue dots show the position of an additional GPS station installed every year (a–d, labelled with the year of operation). Outlines of the calved parts of the glacier in black. White rectangles show the locations of the aerial image and ortho-image in Fig. 6.

## 3.2 GPS measurements

125 In 2019, two rows of five Emlid Reach M+ single-frequency, low cost (about USD 300), differential GPS receivers were installed on stakes drilled into the ice, in close vicinity to the calving front (Fig. 3c), and powered by solar panels. Most of the stations were running from 8 to 25 July 2019, and data were downloaded on a regular basis using Wi-Fi, without touching the devices. Pairs of stations were numbered 1–5, starting from the center towards the southeastern margin, with 'f' (forward) for the station closest to the front and 'b' (backward) further back (Fig. 3c). The spacing between receivers was ≈215 m in 130 across-flow direction and ≈115 m in along-flow direction. A base station was installed on the bedrock next to the glacier to provide a stationary reference point (Fig. 1a). GPS data were processed with RTKLIB using a kinematic mode to obtain the position every second (Takasu and Yasuda, 2009). After differential processing, 95% of the recorded positions were within





**Table 1.** Timeline of events during the 2019 field campaign.

| Date | Event |
| --- | --- |
| 1 July | start of field campaign, timelapse camera installed |
| 8 July | all GPS stations running |
| 14 July | speed-up event |
| 15 July | last time data downloaded from station 3f |
| 20 July | first large calving event (Figs. 3c, S3a-b), during which station 3f was lost |
| 22 July | retrieval of stations 2f and 4f |
| 25 July | retrieval of all remaining stations |
| 29 July | second large calving event (Figs. 3d, S3c-d) and end of field campaign |
|  | timelapse camera retrieved |

10 mm horizontally and 29 mm vertically from the mean values. When taking the 1 h moving average of the vertical position, the maximum deviation of 95% of the recorded positions to the mean position is 6 mm, which we interpret as the vertical

positioning accuracy of our processed GPS data.

One GPS station (3f) calved off during the 20 July event. Data from 3f was downloaded the last time on 15 July. Two of the remaining stations (2f and 4f) were retrieved on 22 July due to the risk of a subsequent calving event. All remaining stations were retrieved by the end of the field campaign on 25 July, four days prior to the second observed major calving event.

To monitor the time evolution of flotation conditions, another GPS station was also installed in between 1b and 2b, at

approximately the same coordinate during each field campaign, (positions shown in Fig. 3). The installation positions in 2015, 2016 and 2019 were located within 20 m, but in 2017 the GPS station was installed approximately 30 m further upstream. The distance to the front varied from 240 m (2015) to 140 m (2019) due to variations in the front position. For more information on this different GPS set-up and processing, we refer to Sugiyama et al. (2015).

### 3.3   UAV photogrammetry

The vertical take-off and landing, fixed-wing uncrewed aerial vehicle (UAV) "WingtraOne" was used to conduct photogrammetrical surveys on 17, 19, 21 and 28 July 2019. The missions consisted of one to four flights of four lines each, parallel to the calving front ≈400 m above the glacier surface, covering 800–3000 m of the terminus of Bowdoin Glacier. The UAV was programmed to take aerial photographs overlapping 90% in flight direction and 75% in across-flight direction, with a ground sampling distance of ≈0.12 m. Ortho-images with 0.5 m resolution were derived by Structure-from-Motion photogrammetry

using Agisoft PhotoScan software. All 10 GPS stations were marked with tarps and served as ground control points (GCPs) for georeferencing. UAV surveys in 2015 and 2017 are described in Jouvet et al. (2017) and van Dongen et al. (2020a).



### 3.4 Satellite images

We constructed an inventory of calving events from 2015–2019, containing all calving events which were large enough to be detected on satellite images. A combination of Landsat 8 Operational Land Imager panchromatic imagery, Sentinel-2 imagery and Sentinel-1 synthetic aperture radar (SAR) imagery is used (Table 2). The visual satellite imagery (Landsat 8 and Sentinel-2) is complemented by Sentinel-1 SAR imagery in case of clouds, such that the image pairs used to outline calving events are maximum six days apart. Spatial resolution of the images varies from 30 m (Landsat 8) to 10 m (Sentinel).

In order to put the scale of detected events into perspective, their area is expressed relatively to the annual ice discharge. To allow direct comparison of calving event sizes between years, we used the annual discharge averaged for 2015–2019 estimated from Sentinel-2 imagery (in combination with the estimate for 2015 derived in Jouvet et al., 2017). From these images, we derive 50 m resolution monthly velocity fields from 2016–2019 by the orientation correlation method with a window size of $40 \times 40$ pixels using the software package ImGRAFT (Messerli and Grinsted, 2015). We estimated an annual ice discharge of approximately $0.9 \pm 0.2 \, \mathrm{km^2 \, yr^{-1}}$, by integrating monthly velocities from 2015–2019 through the flux gate shown in Fig. 7c.

### 3.5 Timelapse images

An automated camera was installed next to the calving front (Fig. 1a), taking daily pictures from July 2014 to July 2017 and hourly pictures during field campaigns in July 2015, 2016, 2017, and 2019. The images were used to further constrain the timing of calving events in the inventory.

### 3.6 Meteorological and ocean data

To investigate tidal forcings, observations of tidal height at Thule Air Base were used, 125 km south of Bowdoin Glacier, provided as a part of the Global Sea Level Observing System network (www.gloss-sealevel.org). These records show the same phase and amplitude as in Bowdoin Fjord (Podolskiy et al., 2016). We obtained hourly air temperatures at Qaanaaq Airport (16 m a.s.l.) located 30 km southwest of Bowdoin Glacier, from the National Oceanic and Atmospheric Administration (http://www.ncdc.noaa.gov/cdo-web/datasets). The temperature dataset was used to constrain the melt season duration, as an indicator for the possibility of hydro-fracturing as a potential driver of calving. The first and last occurrence of at least two consecutive days with positive hourly air temperature are used as proxy for surface melt.

## 4 Results

### 4.1 Elevation change 2014–2017

ArcticDEMs show that most of the glacier front lowered from 2014–2017 (Fig. 4). A particularly strong local lowering is observed after 2015, in the area close to the medial moraine where GPS stations 4b-f are located. There, the lowering rate reached up to $12 \, \mathrm{m \, yr^{-1}}$ between 2015 and 2017. On the bedrock, elevation differences are within $\pm 1$ m, which we interpret as the accuracy of the detected elevation change.





**Table 2.** Overview of the availability of data sets. Field seasons are outlined with vertical lines. Positions of the GPS network are shown in Fig. 3c and d. The position of the single GPS in each year is also shown in Fig. 3. Note that timings are sometimes approximated, as the entire July month is marked as field season each year whereas the campaigns were shorter in 2015–2017.

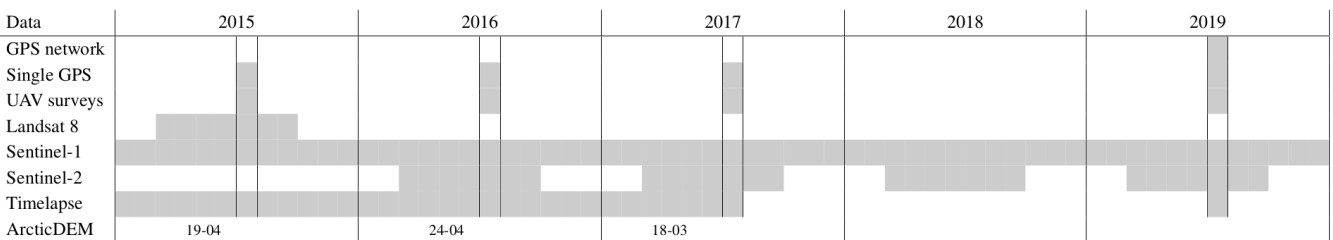

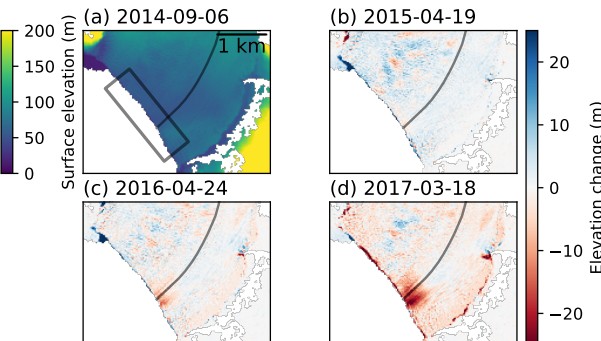

**Figure 4.** Elevation distribution in September 2014 (a) and elevation change from April 2015 to March 2017 (b–d), using ArcticDEM. Colors in (b–d) show the total elevation change relative to September 2014 (negative, red, means lowering). The manually traced location of the moraine is indicated by a transparent black line in all panels, and the location of cropped ortho-images in Fig. 3 is shown by a black rectangle in (a).

## 4.2 Vertical tidal modulation 2015–2019

To investigate the influence of thinning on flotation conditions at the calving front, we analyse the vertical response to semi-diurnal tidal modulation. We calculate the tidal admittance as the range in vertical surface displacement relative to the tidal range (de Juan et al., 2010). We define the range of tidally modulated vertical motion as the elevation difference during low and subsequent high tide. GPS data show that the vertical tidal modulation became significant in 2017 (Fig. 5). In 2019, the calving front was located 55 m closer to the installed GPS than in 2017 (Fig. 3). In both 2017 and 2019, the mean tidal admittance was 0.9%. Also the maximum tidal admittance was equal: 2.6%, observed at 10 July 2017 and 15 July 2019.

## 4.3 Two large-scale calving events in July 2019

Two major calving events were detected from UAV and timelapse imagery, on 20 and 29 July 2019 (Figs. 3c,d and S3). UAV imagery shows that the size of the 20 July event was approximately 700 m by 50 m and was centered close to the moraine

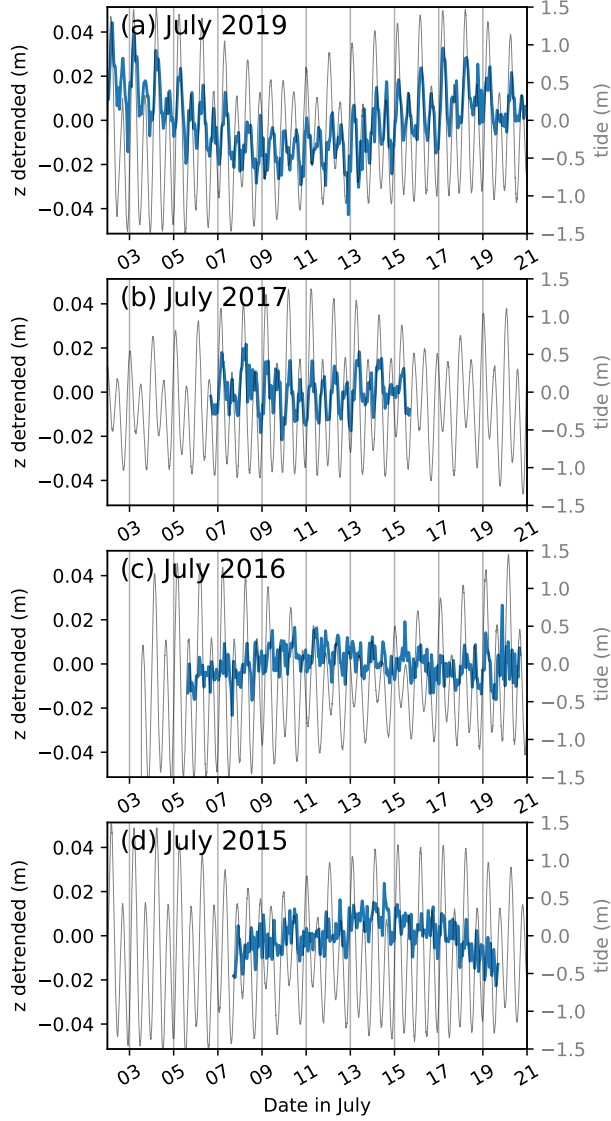

**Figure 5.** Detrended, 1 h moving average vertical position (blue) at the station installed each field campaign (location shown by blue dot in Fig. 3), with the tidal height in the background (black).

(Fig. 3c). Because the field campaign ended on 29 July (Table 1), we have no UAV imagery after the calving event on 29 July. However, by combining UAV and satellite imagery, we find that the 29 July calving event was almost three times as large in area as the 20 July event. It was approximately 1 km by 100 m and widest in the central part of the glacier (Fig. 3d).

In contrast to large-scale calving events observed in July 2015 and 2017, no major surface crevasses were observed prior to calving; neither in the field, nor from aerial imagery. Figure 6a shows an image taken by the UAV, 17 hours before calving

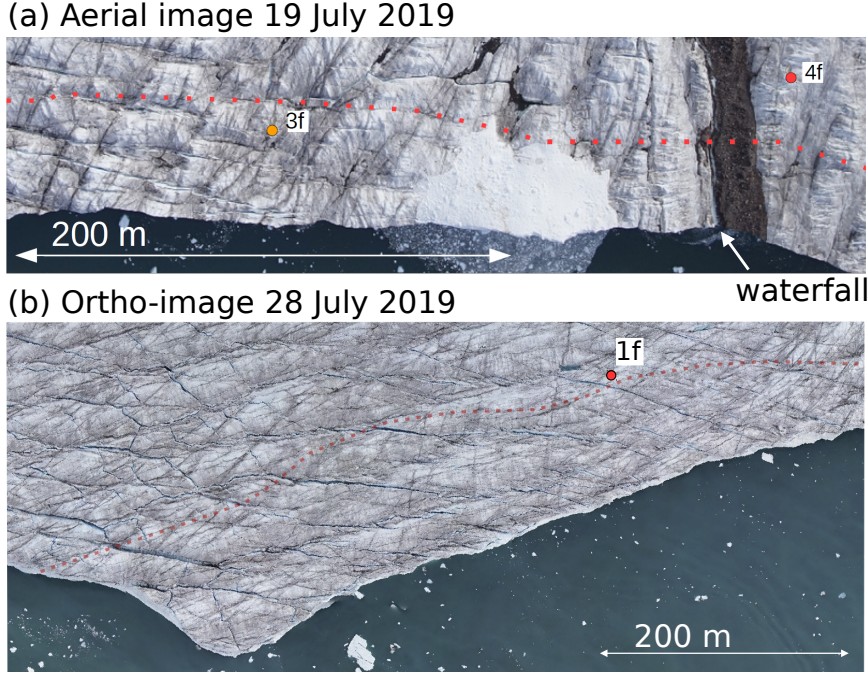

**Figure 6.** (a) UAV aerial image, taken on 19 July 2019, 17 hours before GPS station 3f calved off, but 4f did not calve. (b) Ortho-image 28 July 2019, 14 hours prior to calving. The approximate locations of calving, derived from UAV and satellite imagery, are drawn as dotted red lines. The area of white ice fragments in (a) is discussed in Sect. 5.2. For reference, the location of station 1f at its retrieval on July 25 is shown in (b).

occurred on 20 July, without a crevasse visible at the calving location. Also during the last UAV survey on 28 July, no crevasse was seen on the surface where calving occurred 14 hours later (Fig. 6b). The lack of a precursor crevasse was also evident from the meltwater stream on the moraine, which was at all times observed to discharge into the sea through a waterfall at
the calving front (Fig. 6a). A surface crevasse draining the stream was not observed, contrary to 2015 and 2017 (Jouvet et al., 2017; van Dongen et al., 2020a).

### 4.4  Horizontal flow velocity in July 2019

Both GPS and satellite-derived horizontal velocity are shown in Fig. 7. The velocity in July 2019 is similar in magnitude and tidal modulation as observed in previous years (Sugiyama et al., 2015). Stations 1–4 show very similar velocities in general
(Fig. 7a). As expected, velocity is much lower at stations 5b and 5f, which are located in the slow moving area in the south east (Fig. 7c, see Fig. 3c for positions). The horizontal velocity at stations 1–4 usually reaches a minimum at high tide. A temporary flow acceleration occurred on 14 July, which was most likely induced by surface meltwater input into the subglacial drainage system, resulting in enhanced sliding, as the accelerated flow coincided with the only two consecutive days of temperatures above in 15 °C in Qaanaaq in July 2019. Similar speed-up events were observed after a few days of high air temperatures



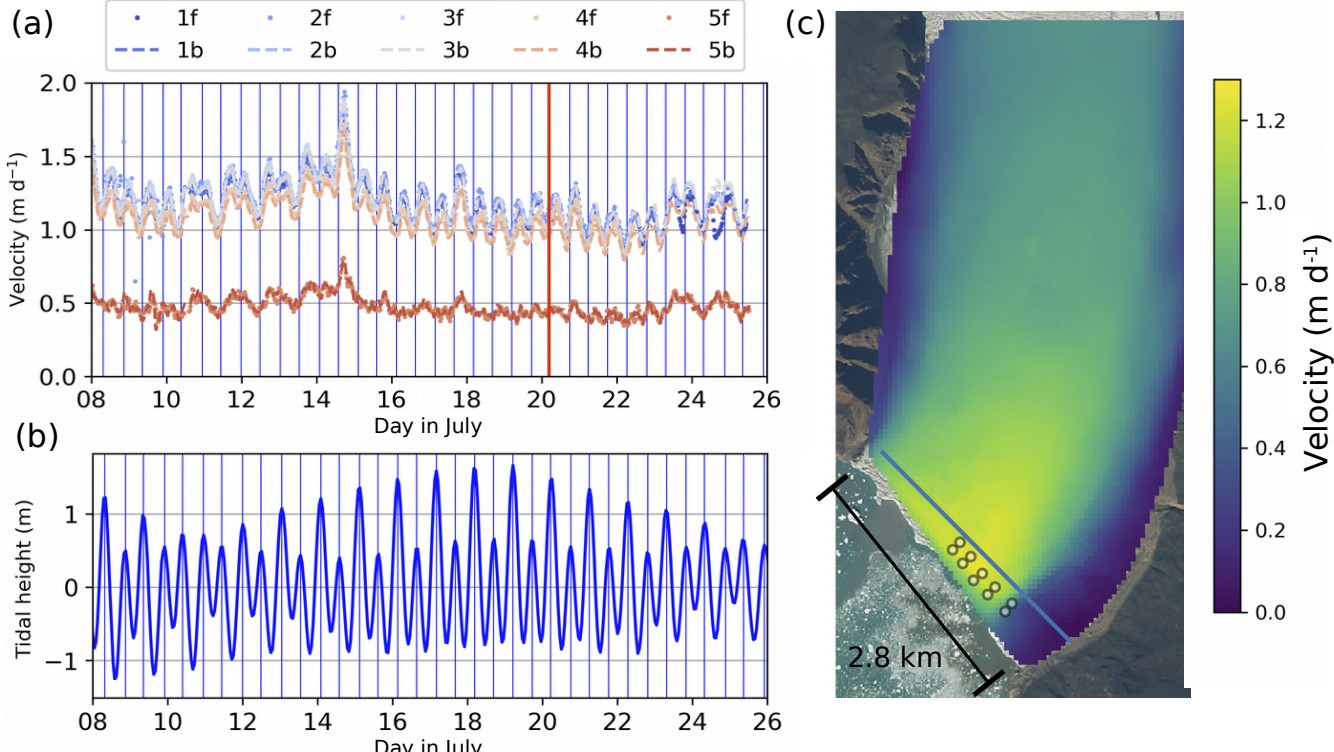

**Figure 7.** (a) GPS-derived horizontal velocity (4 h moving average), (b) tidal height time series and (c) Sentinel-2-derived horizontal velocity by averaging monthly velocities from 12 March – 28 September 2019. Positions of the GPS in (a) are shown in as white dots in (c), for labels see Fig. 3c. Blue vertical lines in (a) and (b) show the timing of high tide, and the red vertical line in (a) the large-scale calving event on 20 July. The blue bar in (c) shows the flux gate used for ice discharge calculation.

(Jouvet et al., 2018) or rainfall (Sugiyama et al., 2015). None of the stations' horizontal velocities are visibly affected by the 20 July calving event (red line, Fig. 7a). Even when velocities are calculated over a shorter time range, or in the 1 s interval positioning data (not shown), no influence of the calving event on horizontal motion is noticeable.

### 4.5 Horizontal strain in July 2019

As a measure of cumulative strain, we calculate the relative distance increase over time $t$ according to: $d(t)/d(t_0) - 1$, where $d$
is the distance between two stations aligned along flow (Fig. 3c) and $t_0$ the initial time. Stations 2b-f and 4b-f show increased extension after the 20 July calving event (Fig. 8b and d), which is likely due to their position closer to the new calving front. A similar increase of extension seems present at the other stations as well, although less clearly. Stations 1b-f show strong horizontal surface compression after 23 July (Fig. 8e), prior to the second major calving event on 29 July.

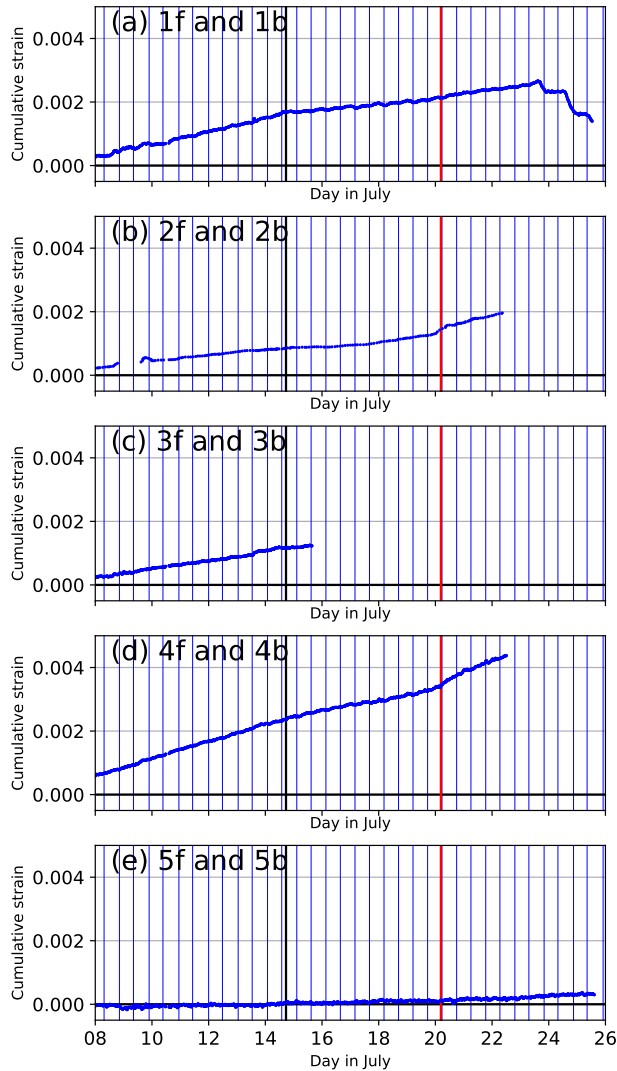

**Figure 8.** Cumulative strain time series for each along-flow station pair (a-e, 1 h moving average), timing of high tide (blue vertical lines), speed-up (black lines) and the 20 July 2019 calving event (red lines, GPS positions in Fig. 3c). Increasing cumulative strain means stretching occurs at the surface, whereas surface compression gives decreasing cumulative strain.

## 4.6   Vertical motion in July 2019

Figure 9 shows the time series of elevation for the GPS stations. Three periods of distinct vertical motion are visible (separated by vertical lines in Fig. 9): 1) prior to the speed-up on 14 July, 2) between the speed-up and calving event on 20 July, and 3) between the calving events on 20 and 29 July. The spatial distribution of the time-averaged vertical motion is displayed in





Fig. 10. Prior to the speed-up, almost all stations west of the moraine move downward (except 1b which shows limited vertical motion), whereas the stations east of the moraine move upward (Fig. 10a).

After the speed-up, only station 1f still moves downward, stations 4f-b move upward quickly and the other stations are initially almost stagnant (Fig. 9b). An abrupt change of vertical motion in response to the speed-up is observed at stations 5f-b, located in the slow moving area (Fig. 9a), whereas the change is rather smooth at the other stations (Figs. 9b-d). A total upward vertical displacement of ≈0.2 m occurred at stations 4f-b between the 14 July speed-up and 20 July calving event (Fig. 9b).

    After calving on 20 July, stations 2f, 3b, 4b and 4f show changes in vertical motion (Fig. 10c). The uplift observed prior to

calving stops (station 4b, Fig. 9b in red), turns into downward motion (4f, Fig. 9b in blue) or downward motion increases (3b, Fig. 9c red and 2f, Fig. 9d blue). The highest rates of downward displacement are observed at 2f and 4f, which are situated closest to the calving event (Fig. 10c). No changes are observed at stations 1b-f, 2b and 5b-f.

### 4.7    Vertical tidal modulation in July 2019

Stations 1–4 show tidal modulation of the vertical position (Fig. 9), whereas this modulation is not visible on stations 5f and 5b,

located in the slow moving area. When subtracting a linear trend, the tidal modulation becomes more clearly visible (Fig. S1). Besides the semi-diurnal tidal modulation, a modulation with a two-week period similar to the spring-neap tidal cycle also seems present, most clearly for stations 3b and 4f-4b. However, as the observational period only spans slightly more than one spring-neap cycle, it is not possible to conclude whether this cycle also modulates vertical motion.

    For most stations, the largest admittance occurred on the evening low tide of 17 July, which is shown in Fig. 11b. The largest

range on 17 July is measured at stations 2f and 4f, which lifted 0.07 m, 3.0% of the tidal range. The range quickly diminishes upglacier, and is only about half as large at 2b and two thirds at 4b. The range is also smaller towards the glacier center (left side of Fig. 11). At station 1f, only 0.04 m uplift is measured, which is slightly more than half the uplift measured at station 2f, 180 m closer to the moraine. These numbers agree well with the additional GPS (Sect. 4.2). On 17 July 2019, a tidal admittance of 1.8% was measured, which is very close to the admittance linearly interpolated from the four nearest stations (1f-b and 2f-b,

where interpolation by triangulation yields 1.9%). At station 3f, for which we have no data on 17 July, the largest admittance is 3.2%, which is obtained on 8 July, as shown in Fig. 11a.

### 4.8    Calving inventory 2015–2019

We manually outlined large calving events visible from satellite imagery from 2015–2019, and calculated the area per event relative to the averaged annual ice discharge. We detected 73 events which form 40–80% of the ice discharge. Detected area

loss between an image pair can reach up to 25% of the annual ice discharge. However, it is impossible to determine from satellite imagery whether these losses are single calving events, even for a minimum time span of one day between image acquisitions. Therefore we have used timelapse imagery (when available) to confirm that these area losses are due to single calving events (Figs. S2, S3). Major loss (≥ 15% of annual ice discharge) in one image pair is detected six times, at least once every year. Together the six largest events form 22% of the estimated ice discharge in the entire period. Figure 12 shows that

large calving events (≥ 5 % of annual ice discharge) are rare outside of the melt season (orange lines), except for four events

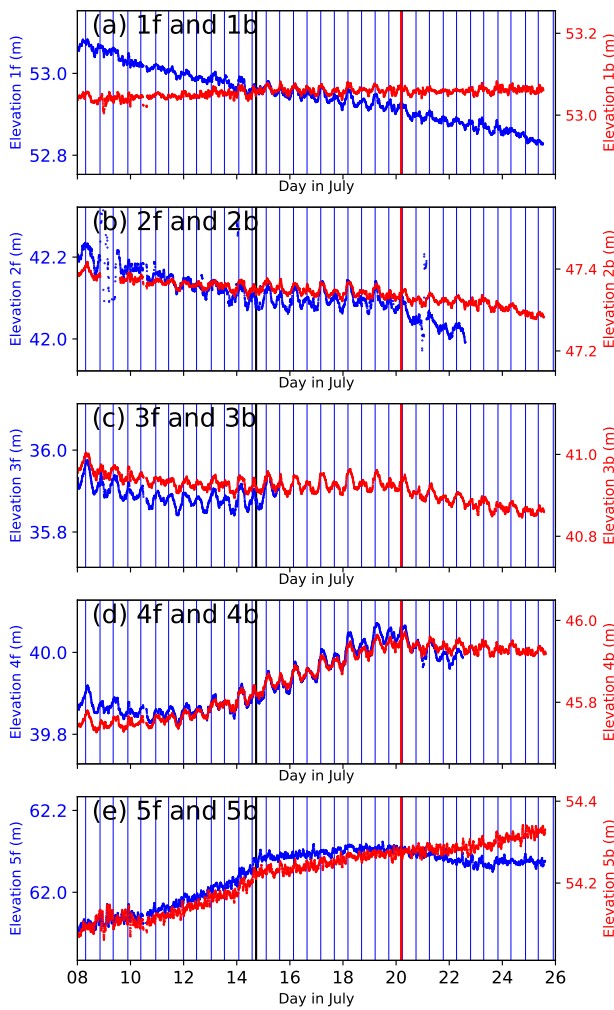

**Figure 9.** GPS-derived elevation time series for each station (a-e, applying a 1 h moving average), timing of high tide (blue vertical lines), the speed-up (black lines) and the 20 July 2019 calving event (red lines). Stations closest to the calving front are shown in blue (1f-5f), stations ≈ 120 m further away in red (1b-5b, see Fig. 3c for positions of GPS stations).

on 4 May 2015, 1 May 2018, 19 March 2019 and one event between 28 December 2019 and 3 January 2020. The relative contribution of calving events outside the melt season to the yearly total is larger in 2018 and 2019 than in the three years before, both in terms of area and number.

## 5 Discussion

When Bowdoin Glacier stabilized to its current position in 2013, its terminus was grounded but very close to flotation (Sugiyama et al., 2015). Field evidence and modelling results showed that two major calving events in July 2015 and 2017 re-

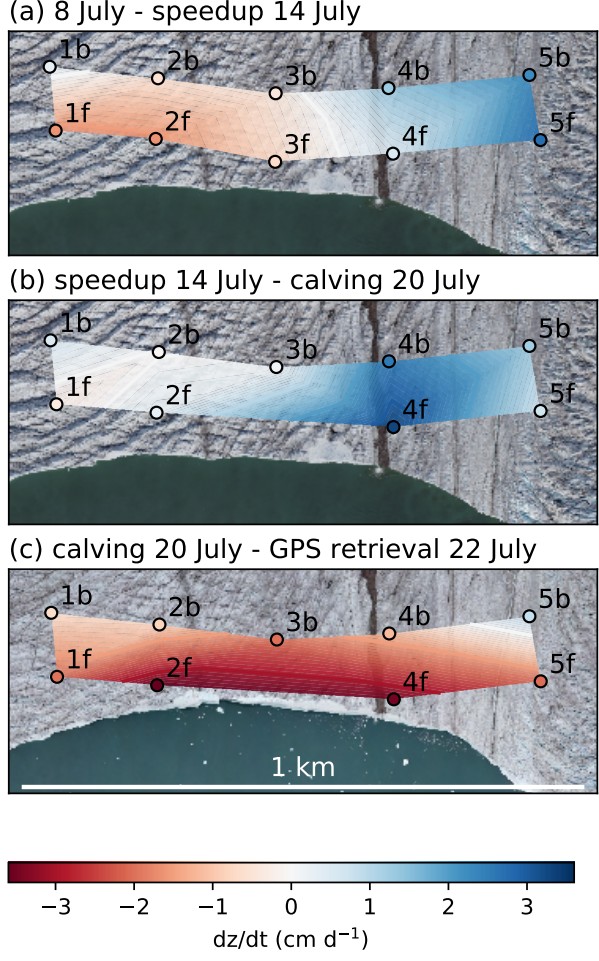

**Figure 10.** GPS-derived elevation change averaged over three time periods indicated above each panel (a)–(c). The observations are interpolated linearly between the stations to visualize the spatial pattern of elevation change. The ortho-images in the background are from 15 July (a,b) and 21 July 2019 (c).

sulted from deep surface crevasses which likely propagated by hydro-fracturing and facilitated by submarine melt-undercutting (Jouvet et al., 2017; van Dongen et al., 2020a, b). Because Bowdoin Glacier has thinned since it stabilized (Fig. 4), we expect its terminus to become ungrounded and thus become subject to buoyant calving. We obtained high-resolution ice motion data from a GPS network in July 2019, which are most suitable to investigate flotation conditions. Therefore, we first discuss why our observations indicate that at least part of the terminus was ungrounded and buoyant calving occurred in July 2019. Then, we combine field and remote sensing data from previous years, and show that ungrounding was measurable from 2017, when calving was still observed to occur through rifting due to extensional stress, and buoyant calving has become more common since 2018.



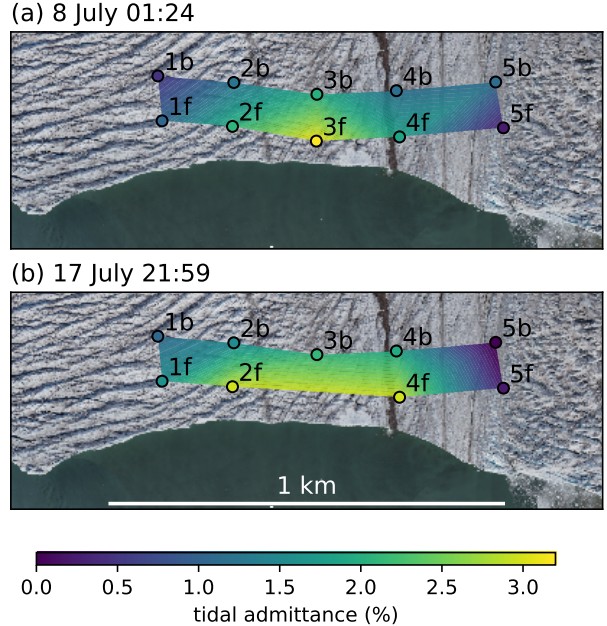

**Figure 11.** GPS-derived tidal admittance (range in vertical position as percentage of the tidal range) for two dates. Tidal admittance is interpolated linearly between the stations for the sake of visualization. Background ortho-images are from 15 July 2019.

## 5.1 Signs of an ungrounded terminus in July 2019

Our GPS measurements indicate in three ways that Bowdoin Glacier's terminus was (at least partially) ungrounded in July 2019:

- vertical motion at Bowdoin's terminus is modulated by tide at stations 1–4, but not at 5f-b (Fig. 9). The magnitude of vertical tidal modulation strongly varies spatially (Fig. 11), which suggests that flotation conditions vary across the front. The absence of tidal modulation in the slow flowing area, where 5f-b are located, indicates that this area is still grounded. The largest tidal modulation is observed close to the moraine, thus this area is presumably subject to largest buoyant forces.

- there is no short-lived horizontal velocity response to the calving on 20 July. This suggests that the calved block did not contribute significantly as a resistive force to the near-terminus force balance. Basal drag caused by this block was likely small, which could be the result of the terminus being at flotation there.

- a change in vertical motion after the 14 July speed-up is most abrupt at 5f and 5b, which show fast upward motion during the speed-up and reduced upward motion afterwards (Fig. 9e). In other years, GPS measurements on Bowdoin Glacier also showed upward motion during fast flow (relative to a downward trend), which has been interpreted as basal separation caused by elevated basal water pressure (Sugiyama et al., 2015). Basal separation is only relevant in areas





where the glacier is grounded, and basal water pressure is governed by meltwater input to the bed. Therefore, the vertical
motion during the speed-up event supports our assumption that the slow flowing area where stations 5f-b are located
is still grounded. The absence of change in vertical motion related to the speed-up at stations 1–4 suggests that basal
separation did not occur there, as would be expected if this part of the terminus was already afloat.

Interpretation of the observed tidal modulation of horizontal velocity in relation to the terminus' flotation condition is more
complex. We observe lowest horizontal velocities at high tide. This contradicts the expected behaviour if the basal hydrological
system would be well connected to the ocean. In this case, increased basal water pressure during high tides would reduce basal
drag and enhance ice motion as has been observed on ice shelves (Brunt, 2008). However, velocity minima at high tide have
also been observed at fast-flowing tidewater glaciers in Alaska (Walters and Dunlap, 1987; O'Neel et al., 2001) and Greenland
(Davis et al., 2007; de Juan et al., 2010; Podrasky et al., 2014). Slow-down at high tide suggests that the dominant effect of
tides is the back-pressure at the calving front (Thomas, 2007). Since similar tidal modulation of horizontal velocity has been
observed for a tidewater glacier with regions of floating ice (Podrasky et al., 2014), the observed slow-down at high tide does
not exclude the possibility of a (partially) ungrounded terminus.

## 5.2 Buoyant calving in July 2019

Unlike the deep surface crevasses which formed prior to calving in 2015 and 2017 (Fig. 2), no precursor crevasse was visible
in the field prior to the major calving events in July 2019. This suggests that the events in July 2019 were not triggered by
extensional stress at the glacier surface, but as a result of basal crevasse propagation due to buoyant flexure instead.

Our GPS observations are consistent with buoyant calving as observed at Helheim Glacier by Murray et al. (2015). They
observed uplift and rotation of the front, creating basal crevasses, and forming a surface depression above the basal crevasses,
which they call a "flexion zone". The basal crevasses induced buoyant calving of the entire area downglacier of the flexion
zone. Such behaviour is particularly clearly visible at our stations 1b-f (Fig. 9e). Station 1f lies on the outline of the 29 July
calving event and thus in the flexion zone. Downward motion is observed at 1f in the entire measurement period, whereas no
downward motion is observed at station 1b, which is located only 115 m upstream. The horizontal compression at stations 1b-f
could be related to buoyant calving on 29 July as well. During the afternoon high tide at 24 and 25 July, sudden longitudinal
compression is observed between stations 1b-f (Fig. 8e). On such short timescales, ice is expected to behave as a brittle-elastic
material (Benn et al., 2017). Therefore, we interpret basal crevasse formation as the most likely cause of the longitudinal
compression at the surface. The highest observed compression at high tide is also favorable for this interpretation, as buoyant
forces which trigger basal crevasse formation increase at high tide.

On the other hand, the uplift at 4b-f prior to calving on 20 July is not entirely consistent with Murray et al. (2015), since they
observed that the entire zone which lifted up subsequently calved off. However, James et al. (2014) observed similar behaviour
on Helheim Glacier, using daily DEMs derived from terrestrial photographs, showing that the uplifting part of the terminus did
not calve at once, but broke off in multiple stages. Stations 2f, 3b and 4b-f, which were all close to the area which calved on 20





July, showed downward motion after calving (Fig. 10c). If the part of the terminus that broke off on 20 July was indeed afloat before calving, the buoyant forces on the terminus became smaller after calving, which could explain the downward motion.

Another sign of buoyant calving is visible on Figure 2a. The pile of white, clean ice fragments, located close to the moraine,
was already present when the field campaign started on 1 July 2019. We speculate that this feature formed during a previous calving event. During detachment of an iceberg by buoyant calving, buoyant forces can push the iceberg above the calving front. A portion of the iceberg could then have fragmented and fallen on the glacier surface immediately behind the calving front. In the field campaigns prior to 2019, a similar feature was never observed.

### 5.3 Glacier thinning since 2014 and (partial) flotation

Bowdoin Glacier has experienced further thinning since the front stabilized to its new position in 2013. The region of strongest thinning (dark red in Fig. 4) coincides with the region of largest observed vertical tidal response in 2019 (yellow in Fig. 11). A surface depression in this region is clearly visible in the field, with the lowest ice cliff located at the moraine. Because the strongest thinned area also coincides with the shear zone, dynamic thinning likely caused this surface depression. Alley et al. (2019) also observed that high shear resulted in surface depressions and subsequently basal channels formed beneath the
depressions because buoyant forces push the locally thinned ice upward. Plumes were observed to originate from these basal channels, which amplified melt locally and thinned these areas further. This proposed shear thinning mechanism agrees with our observations at Bowdoin Glacier, where a plume has been clearly visible on the sea surface close to the moraine during field campaigns in 2015–2017 (Jouvet et al., 2017, 2018; van Dongen et al., 2020a) and in early July during the field campaign in 2019 (picture not shown).

Significant vertical tidal modulation was first observed by GPS in July 2017 (Fig. 5). The mean and maximum tidal admittance were similar in July 2017 and 2019, although the GPS was located ≈55 m further away from the front in 2017 (Sect. 4.2). Because there was only one GPS installed close to front in previous years, outside of the area which showed largest tidal response in 2019, it is difficult to conclude whether vertical tidal modulation changed since 2017.

### 5.4 A changing calving mechanism

There is a large variation in the cumulative area loss by calving detected in each year in our satellite-derived calving inventory (Fig. 12), which has two possible explanations. Firstly, the low cumulative percentage of calved area reported for 2017 (44%) is likely due to the front advance which occurred in 2017 (Fig. S4), which suggests that the calving rate was smaller in 2017, such that the averaged annual ice discharge for 2015–2019 overestimates the frontal ablation in 2017. Similarly, the front retreat in 2016 and 2019 can explain the high reported cumulative area loss relative to annual discharge in those years (≈80%). Secondly,
our inventory aims to study large-scale calving events. Small calving events cannot be detected from low resolution satellite imagery. Minowa et al. (2019) analysed calving-generated tsunami waves and detected 11 calving events per day from 11 to 21 July 2015 and 15 per day from 15 to 30 July 2016, whereas we did not detect any calving events in these periods. As such, variations in area loss by small-scale calving events could also explain part of the variation of undetected area loss in each year.





Our large-scale calving event inventory is consistent with a changing calving mechanism from hydro-fracturing and undercutting-
induced calving towards buoyant calving. Since the precursor surface crevasse in 2017 was not visible on satellite imagery,
we cannot use satellite imagery to directly conclude whether precursor-free calving events happened in previous years as well.
However, the timing of observed large-scale calving events may help to distinguish plausible calving mechanisms. Initiation
and propagation of surface crevasses are promoted by undercutting and hydro-fracturing, which depend on external forcings as
surface and submarine melt. Therefore, they are expected to follow a seasonal pattern (e.g. Medrzycka et al., 2016). Contrarily,
buoyant calving does not depend as directly on seasonal variations.

Satellite imagery shows that major calving events ($\approx$25% of the estimated yearly ice discharge) occurred on 1 May 2018
and 19 March 2019. These calving events are unlikely induced by hydro-fracturing of surface crevasses as it is improbable that
surface melt occurred. Furthermore, warm Atlantic water is not available in winter since the glacier flows into a fjord shallower
than 200 m, which means that substantially large submarine melting capable of destabilizing the front by undercutting is
expected to start once fjord circulation is increased by subglacial meltwater discharge (Sakakibara and Sugiyama, 2020).

A major calving event ($\approx$20%) was also observed on 4 May 2015, prior to surface melt, without a surface crevasse visible on
timelapse imagery (Fig. S2a-b). New sonar data collected close to the front (van Dongen et al., 2020a, not included in Fig. 1b)
shows that the fjord bed topography is seaward sloping at the current front position, before sloping upward towards the bed
bump where the front was located in 2007. The winter advance therefore relocated the front to a deeper part of the fjord in
2015, which could have induced temporary flotation of the front and buoyant calving. A similar mechanism of buoyant calving
triggered by winter advance can also have caused the 12% calving event which occurred around 31 December 2019 (Fig. 12).

The relative contribution of calving outside of the melt season has increased since 2018 (Fig. 12). These calving events were
most likely not melt-driven but mechanically-driven, in response to buoyant-flexure (James et al., 2014). Although buoyant
calving may have occurred in 2015 already, a change seems to have occurred towards buoyant calving as the dominant calving
style since 2018. However, a longer time span of high resolution ice motion data is necessary to confirm a switch towards
buoyant calving. On Columbia Glacier (Alaska), Walter et al. (2010) found that calving behaviour changed from rather steady
discharge of small icebergs to episodic release of large icebergs when the glacier's tongue became afloat. We do not see a
similar trend in the calving inventory at Bowdoin Glacier, which could be due to the different flotation conditions. Whereas the
entire last 2 km of Columbia Glacier became afloat, the eastern marginal area at Bowdoin Glacier is still grounded.

## 375   6   Conclusion and outlook

This study investigated calving processes at Bowdoin Glacier, Northwest Greenland, by combining in situ GPS measurements
and remote sensing data from 2015–2019. Previous studies showed that in July 2015 and July 2017, large-scale calving events
on Bowdoin Glacier were triggered by surface crevasses, facilitated by melt-undercutting and propagated by hydro-fracturing
(Jouvet et al., 2017; van Dongen et al., 2020a, b). Our new observations suggest that the calving behaviour of Bowdoin Glacier
may have changed since then, towards buoyant calving in response to dynamic thinning. We found several signs that part of
the terminus has ungrounded since 2017:



- semi-diurnal tide-modulated vertical motion is observed since July 2017, after years without significant tidal uplift,

- in July 2019, strongest tidal modulation is observed in the shear zone, coinciding with the area of strongest thinning,

- whereas GPS measurements suggested basal separation occurred during speed-ups in 2015, signs of basal separation are
no longer visible during a speed-up in 2019 for most GPS stations, which suggests this area was not grounded.

Furthermore, our data suggests that buoyant calving recently occurred:

- major calving events in July 2015 and 2017 were announced by deep surface crevasses, whereas no precursor was observed prior to two large calving events in July 2019,

- GPS measurements during and prior to calving in July 2019 are consistent with a calving mechanism dominated by
buoyant flexure, resulting in upward rotation of ice that has reached flotation,

- our inventory shows that large-scale calving events became more frequent outside of the melt season since 2018.

As Bowdoin Glacier's terminus is located on an overdeepened bedrock, further thinning may increase the frequency of buoyant calving events and induce glacier retreat until the terminus reaches the next bedrock bump, approximately 1.2 km upstream its current position (Fig. 1b). To get more insight on Bowdoin Glacier's sensitivity to sustained thinning and external
forcings, one would need to perform numerical simulations that resolve spatial and temporal variations of flotation conditions, similar to Todd et al. (2019). Dynamic thinning is commonly observed on outlet glaciers of the Greenland and Antarctic Ice Sheets (Pritchard et al., 2009; IMBIE Team, 2019) and there is a large potential for future dynamic thinning (Price et al., 2011; Felikson et al., 2017). A dynamic thinning-induced transition to buoyant calving may thus be expected for many glaciers worldwide. Our study highlights the importance of representing the full range of calving mechanisms observed in nature to
yield reliable projections of future ice sheet mass loss by retreat of outlet glaciers. Although high-resolution models have recently been developed that are capable of reproducing different calving styles (e.g. Benn et al., 2017), the major challenge will be to translate these findings into calving laws that improve long-term projections of ice-sheet mass loss.

*Data availability.* The UAV data collected in 2015 are available at Jouvet et al. (2019). The GPS network data and 2019 UAV data will be published in ETH Zurich's Research Collection upon publication of this manuscript.

*Author contributions.* EvD, GJ, MF and FW designed the study. MF, GJ, FW and SS organized the field campaign at Bowdoin Glacier. GPS stations were designed and prepared by GJ, AB, EvD, and FL. EvD, FL and AB installed the GPS network with support from SS and EP, and the data were processed by GJ. SS installed the additional GPS in 2015-2019 and was responsible for processing its data. JS processed Copernicus Sentinel-2 data with Sentinelflow (doi:10.5281/zenodo.1774659) and SL processed Sentinel-1 imagery. EvD analysed the data and drafted the manuscript with support from GJ, SS, EP, MF, DB and FW, and all authors contributed to the final version.



*Competing interests.*  The authors declare that they have no conflict of interest.

*Acknowledgements.*  We thank the members of the field campaigns on Bowdoin Glacier, in particular Joachim Wassermann who assisted maintaining the GPS network in 2019. We wish to acknowledge Toku Oshima and Kim Petersen for assistance with field activities in Qaanaaq. We also thank Thomas Wyder for preparing the GPS devices for installation on the glacier surface and Amaury Dehecq for assistance with DEM coregistration. This research is part of the Sun2ice project (ETH Grant ETH-12 16-2), supported by the Dr. Alfred and Flora Spälti

and the ETH Zurich Foundation. Fieldwork was funded by the Swiss National Science Foundation (SNSF), grant 200021-153179/1, the SPI Exploratory Grants 2018 awarded to Fabian Walter and Guillaume Jouvet, and the Japanese Ministry of Education, Culture, Sports, Science and Technology through the GRENE Arctic Climate Research Project and the Arctic Challenge for Sustainability (ArCS) project. The salary of Fabian Walter was financed by the SNSF via Grant PP00P2_183719. Douglas Benn was funded by NERC grant NE/P011365/1 (CALISMO: Calving Laws for Ice Sheet Models).



**Figure 12.** Time series of the area per calving event inferred from satellite image pairs (red, left $y$-axis) and the cumulative annual area loss by calving (blue, right $y$-axis), both relative to estimated annual discharge of $0.9 \; \mathrm{km}^2$. Horizontal red bars show when the date of calving is unknown. Vertical orange lines show the first and last two consecutive days with positive hourly temperatures. The dashed orange line in 2017 indicates a separate 2-day peak with positive temperatures in December. The source of satellite imagery is indicated in the top of each panel. The average calved area per event ($A_{av}$), number of events ($N_{tot}$) and the relative contribution of calved area and number of events in winter ($A_{winter}$, $N_{winter}$) are given for each year.



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
