# Peer review of "Thinning leads to calving-style changes at Bowdoin Glacier, Greenland"

_The Cryosphere, 2020_

## Referee Comment (RC1) · Basile de Fleurian (Referee) · 19 Nov 2020

**1 General comments**

This study highlights the evolution of the Bowdoin glacier calving style towards buoyant calving. The shift in calving style is exposed through a multi method monitoring using both field based and remotely sensed data.

The paper first goes through the different dataset that have been acquired in great details before analysing those results to assess the timing of the ungrounding of the glacier. The analysis is then discussed in term of calving style before concluding on the effect on mass loss that can be measured from space.

[Figure]

This study is clear and well written, I have just a few general comments and some minor points that are listed in Section 2.

In Section 4.1 I don't completely agree with the analyse of the thickness data evolution. From the presented figure it seems to me that the front thickened from 2014 to 2015 before starting to thin, isn't that the case? Did you use a global variable here that shows lowering during 2014?

Regarding the calving quantification, the calving area is reported to the annual discharge which is given in $km^2yr^{-1}$ and not $km^3yr^{-1}$ as one would expect. I suppose that this is due to the fact that the annual displacement has not been integrating over the ice thickness but that should be stated.

In Section 5.1 the effect of the tides are analysed to assess the flotation of the ice front. Have you considered to look at the strain measurements in correlation to tides to strengthen this case. It seems that at least on station 1 there is a signal in the strain that looks to be in phase with the tides with a compression regime at high tide. I also wonder if it would not be better to plot the instantaneous values rather than cumulative ones on Figure 8 to help with readability.

**2  Specific comments**

Bellow is a list of more specific comments throughout the manuscript given with line and page number:

- Line 19: Projections is missing a capital.

- Line 29: I am not sure that I get the meaning of "large-scale" here.

- Line 61: It is the glaciers fronts that advance and retreat depending on the season, not the glaciers themselves.

- Line 71: Replacing "the submarine melt rate increases in summer in shallow fjords" by "in the shallower fjords submarine melt rate increases in summer" would make that sentence clearer in my opinion.

- Line 75: Shouldn't King (2020) be added to the IMBIE team (2018) reference here.

- Line 212: The end of this sentence is a repetition from the one above and could be omitted.

- Line 230: "9b" should be "9d" twice on this line.

- Line 231: "9d" should be "9b".

- Line 249: Is it annual "ice discharge" that is meant here?

- Line 305: "9e" should be "9a".

- Line 307: A reference to Fig. 8a should be added.

- Line 309: "8e" should be "8a".

- Line 313: A reference to Fig. 8d should be added.

King, M. D., Howat, I. M., Candela, S. G., Noh, M. J., Jeong, S., Noël, B. P. Y.and Negrete, A. 2020. Dynamic ice loss from the greenland ice sheet driven by sustained glacier retreat. Communications Earth & Environment. 10.1038/s43247-020-0001-2

IMBIE team. 2018. Mass balance of the Antarctic Ice Sheet from 1992 to 2017. Nature. 10.1038/s41586-018-0179-y10.1038/s41586-018-0179-y

---

## Referee Comment (RC2) · Anonymous Referee #2 · 24 Nov 2020

In this paper van Dongen et al. undertake a detailed study of Bowdin Glacier, NW Greenland, utilising an impressive combination of remote sensing and field data. Utilising these data, they argue that there has been a change in calving style from submarine melt driven calving to buoyant flexure driven calving. Overall I'm convinced by the arguments presented in the paper, though have a few minor points outlined below. However, the article is generally very well written, and the authors should be congratulated on pulling together such a nice range of data.

L185-6 – a bit more explanation as to why figure 5 shows the vertical tidal modulation being significant would be beneficial. At present it's unclear why results are significant in panel b but no in panel c/d

L248 (& L155) – it's mentioned earlier on that Sentinel 1 data are analysed – how are

these processed to ensure comparability/homogeneity with optical data?

L263 – I would change "expect" to a phrasing that is a lot less definitive. Unless you have thickness information to demonstrate that thinning is sufficient to lead to the terminus becoming fully ungrounded, it would be more appropriate to say that the terminus is getting closer to floatation

L260 – I quite like the your intro to the discussion (just having now deleted my comment saying it wasn't needed!). For a data heavy paper like this it lines up and signposts the main points in the discussion nicely.

Section 5.1 – compelling case that the terminus was partially ungrounded.

L299 – are these crevasses visible from satellite or available time lapse imagery? I'm not 100% convinced that the photographs provided do actually give definitive evidence of calving being propagated from extensional rifts in 2015/17. It may be worth reiterating evidence from previous studies backing submarine melt undercutting as the mechanism. However, I agree that the GPS observations point towards a buoyant flexure style of calving in 2019.

L328-331 – figure 4 is very small, which makes clear identification of the small scale variability of thinning and also the location relative to the moraine. I suggest this figure be made larger and an extra panel showing a satellite image of the glacier (with centreline marked) that would allow clear identification of these. Regarding the dynamic thinning hypothesis I'm not entirely sure what you mean by this with respect to the shear zone. Similarly with the mechanism for shear forming basal channels beneath the depressions. This section would benefit from clarification.

Section 5.4 – showing panels for each year of the glacier terminus positions used to derive the calving event sizes would be useful to compliment figure 12, showing where and how each calving event occurred. It would also aid the discussion in this section (5.4) which is a bit tricky to follow. As it is, figure 1a showing ice front info is not

especially useful/relevant to the study as it is annual and goes back to 1973, whereas the period under investigation is 2015-19.

More generally, I would say that the observations presented do not represent clear cut evidence of a transition from one calving style to the next. For example, given the stochastic nature of calving I would be a little cautious in ascribing too much meaning to a few individual events that happen pre/post initiation of melt, even though they may be substantial. What is more convincing for me is the combination of the previous studies (Jouvet et al., 2017; van Dongen et al., 2020a,b) identifying that large scale calving events occurred due to hydrofracture/melt undercut, the arguably circumstantial evidence presented here, and evidence for recent ungrounding combined with GPS data showing evidence of calving by buoyant flexure. I think the discussion would benefit if these lines of evidence were linked more strongly.

---

## Author Comment (AC1) · 10 Dec 2020

**Reply to RC1**

We thank the referee for the time and effort spent on reviewing our manuscript. A point-by-point reply (in blue) to each comment is given below.

**1 General comments**

This study highlights the evolution of the Bowdoin glacier calving style towards buoyant calving. The shift in calving style is exposed through a multi method monitoring using both field based and remotely sensed data.

The paper first goes through the different dataset that have been acquired in great details before analysing those results to assess the timing of the ungrounding of the glacier. The analysis is then discussed in term of calving style before concluding on the effect on mass loss that can be measured from space.

This study is clear and well written, I have just a few general comments and some minor points that are listed in Section 2.

Thank you.

In Section 4.1 I don't completely agree with the analyse of the thickness data evolution. From the presented figure it seems to me that the front thickened from 2014 to 2015 before starting to thin, isn't that the case? Did you use a global variable here that shows lowering during 2014?

Thank you for pointing out this ambiguity. Indeed, Fig. 4b suggests that the front thickened from 2014 to 2015, but this is a result of our choice to use the ArcticDEM strip of 2014-09-06 as reference, which is then compared to the ArcticDEMs acquired in March and April 2015 – 2017, such that the winter accumulation results in a thickening between September 2014 and April 2015. Unfortunately, no ArcticDEM is available for March-April in 2014, but we propose to change the reference to the ArcticDEM of 2013-04-04, such that all elevation models are from approximately the same time of the year. This shows that the front is indeed thinning during the entire period.

[Figure]

[Figure]

Regarding the calving quantification, the calving area is reported to the annual discharge which is given in $km^2$ $yr^{-1}$ and not $km^3$ $yr^{-1}$ as one would expect. I suppose that this is due to the fact that the annual displacement has not been integrating over the ice thickness but that should be stated.

Yes, this is correct, the displacement is only integrated horizontally, not over the ice thickness, which we will state in the revised manuscript.

In Section 5.1 the effect of the tides are analysed to assess the flotation of the ice front. Have you considered to look at the strain measurements in correlation to tides to strengthen this case. It seems that at least on station 1 there is a signal in the strain that looks to be in phase with the tides with a compression regime at high tide. I also wonder if it would not be better to plot the instantaneous values rather than cumulative ones on Figure 8 to help with readability.

Yes, we have tried plotting instantaneous values of strain rather than cumulative, which indeed highlights the tidal modulation of strain with compression at high tide, but it makes it more difficult to see the longer term changes in strain related to the calving events. Therefore we chose to show the cumulative strain instead. We find it difficult to draw conclusions concerning the flotation condition from the compression regime at high tide, because this could also be related to tidal modulation resulting from the water pressure forcing at the ice cliff, not necessarily from buoyant forces below the ice. In that sense, we do not think the observed tidal modulation of strain can be conclusive to support either flotation or grounded conditions for this specific case.

**2 Specific comments**
Bellow is a list of more specific comments throughout the manuscript given with line and page number:
• Line 19: Projections is missing a capital.

Thanks, we will fix this in the revised manuscript.

• Line 29: I am not sure that I get the meaning of "large-scale" here.

Here, 'large-scale' is also meant relative to the glacier size, hence velocity gradients which are present on a large portion of the calving front. In the revised manuscript, we remove 'large-scale' from L29 as your comment made us realise that it can be confusing after introducing 'large-scale calving events' in the preceding paragraph.

• Line 61: It is the glaciers fronts that advance and retreat depending on the season, not the glaciers themselves.

We will rephrase to 'most tidewater glacier termini on Greenland advance in winter'

• Line 71: Replacing "the submarine melt rate increases in summer in shallow fjords" by "in the shallower fjords submarine melt rate increases in summer" would make that sentence clearer in my opinion.

Thanks, we will rephrase this.

• Line 75: Shouldn't King (2020) be added to the IMBIE team (2018) reference here.

Yes, we will add this.

• Line 212: The end of this sentence is a repetition from the one above and could be omitted.

We will omit the last part of line 212.

• Line 230: "9b" should be "9d" twice on this line.

We will fix this.

• Line 231: "9d" should be "9b".

We will fix this.

• Line 249: Is it annual "ice discharge" that is meant here?

Yes, we will specifically mention this.

• Line 305: "9e" should be "9a".

We will fix this.

• Line 307: A reference to Fig. 8a should be added.

We will add a reference to 9a (8a concerns the strain, not the elevation mentioned in this sentence).

• Line 309: "8e" should be "8a".

We will fix this.

• Line 313: A reference to Fig. 8d should be added.

We will add a reference to 9d (8d concerns the strain, not the elevation mentioned in this sentence).

King, M. D., Howat, I. M., Candela, S. G., Noh, M. J., Jeong, S., Noël, B. P. Y.and Negrete, A. 2020. Dynamic ice loss from the greenland ice sheet driven by sustained glacier retreat. Communications Earth & Environment. 10.1038/s43247-020-0001-2

IMBIE team. 2018. Mass balance of the Antarctic Ice Sheet from 1992 to 2017. Nature. 10.1038/s41586-018-0179-y10.1038/s41586-018-0179-y

---

## Author Comment (AC2) · 10 Dec 2020

**Reply to RC2**

We thank the referee for the time and effort spent on reviewing our manuscript. A point-by-point reply (in blue) to each comment is given below.

**Anonymous Referee #2**

In this paper van Dongen et al. undertake a detailed study of Bowdoin Glacier, NW Greenland, utilising an impressive combination of remote sensing and field data. Utilising these data, they argue that there has been a change in calving style from submarine melt driven calving to buoyant flexure driven calving. Overall I'm convinced by the arguments presented in the paper, though have a few minor points outlined below.

However, the article is generally very well written, and the authors should be congratulated on pulling together such a nice range of data.

Thank you.

L185-6 – a bit more explanation as to why figure 5 shows the vertical tidal modulation being significant would be beneficial. At present it's unclear why results are significant in panel b but no in panel c/d

Thank you for pointing this out. This was only a visual interpretation, for which 'significant' was a too strict formulation. We have now calculated correlation coefficients for a linear regression between the tidal height and ice elevation for each year, yielding Pearson r correlation coefficients of 0.27 for 2015, 0.29 for 2016, 0.71 for 2017 and 0.58 for 2019 (all significant). We will add this information to the revised manuscript:

"*GPS data show that the vertical tidal modulation increased since 2017 (Fig. 5), with a stronger correlation between detrended elevation and tidal height in 2017 and 2019 (r = 0.64) than in 2015 and 2016 (r=0.28).*"

L248 (& L155) – it's mentioned earlier on that Sentinel 1 data are analysed – how are these processed to ensure comparability/homogeneity with optical data?

We will add the following information on the processing of Sentinel-1 data:

"*Sentinel-1 data were downloaded using the Google Earth Engine (Gorelick et al., 2017) which provides backscatter coefficients orthorectified using the ASTER DEM to the same coordinate system as Sentinel-2 imagery (UTM 19N). To minimize incidence-angle dependent effects and also artefacts due to inaccuracies in the DEM, we processed data from a single orbit (026, descending). To correct for sub-pixel shifts between the orthorectified images, we coregistered them to a common reference (intensity average of all scenes) using intensity cross correlation (Strozzi et al., 2002).*"

L263 – I would change "expect" to a phrasing that is a lot less definitive. Unless you have thickness information to demonstrate that thinning is sufficient to lead to the terminus becoming fully ungrounded, it would be more appropriate to say that the terminus is getting closer to floatation.

We have rephrased '*the terminus is now closer to flotation and thus may have become subject to buoyant calving.*'

L260 – I quite like the your intro to the discussion (just having now deleted my comment saying it wasn't needed!). For a data heavy paper like this it lines up and signposts the main points in the discussion nicely.

Thank you.

Section 5.1 – compelling case that the terminus was partially ungrounded.

L299 – are these crevasses visible from satellite or available time lapse imagery? I'm not 100% convinced that the photographs provided do actually give definitive evidence of calving being propagated from extensional rifts in 2015/17. It may be worth reiterating evidence from previous studies backing submarine melt undercutting as the mechanism. However, I agree that the GPS observations point towards a buoyant flexure style of calving in 2019.

Yes, these crevasses are visible from time lapse but the crevasse in 2017 was not visible from satellite (as mentioned on L350 of the discussion paper). We will move the first paragraph (L299-301) to Sect. 5.4, and will reiterate the evidence from previous studies there, as explained in more details in the reply to your last comment.

L328-331 – figure 4 is very small, which makes clear identification of the small scale variability of thinning and also the location relative to the moraine. I suggest this figure be made larger and an extra panel showing a satellite image of the glacier (with centreline marked) that would allow clear identification of these. Regarding the dynamic thinning hypothesis I'm not entirely sure what you mean by this with respect to the shear zone. Similarly with the mechanism for shear forming basal channels beneath the depressions. This section would benefit from clarification.

We will increase the size of Figure 4, and add a reference to the satellite image in Fig. 1a on which the moraine is marked. For clarification purposes, we suggest to rephrase the section as follows: '*Because the strongest thinned area also coincides with the shear zone, the large velocity gradients likely caused this surface depression by dynamic thinning. A similar case was reported by Alley et al. (2019), who used satellite data to show that high shear resulted in surface depressions. Their observations suggest that the locally thinned ice was subsequently pushed upward by buoyant forces, and basal troughs formed beneath the depressions. Plumes were observed to focus into these basal troughs, which amplified melt locally and thinned these areas further. This proposed shear thinning mechanism agrees with our observations at Bowdoin Glacier ...*'

Section 5.4 – showing panels for each year of the glacier terminus positions used to derive the calving event sizes would be useful to compliment figure 12, showing where and how each calving event occurred. It would also aid the discussion in this section (5.4) which is a bit tricky to follow. As it is, figure 1a showing ice front info is not especially useful/relevant to the study as it is annual and goes back to 1973, whereas the period under investigation is 2015-19.

We will adapt Fig. 1A, to show the front positions of 1973, 2008, 2013 and 2015-2020, as we agree that it would be helpful to show the front positions of 2015-2020 here. However, we do find it relevant to show the retreat history of the glacier, although in less detail than in the discussion paper to have more focus on the recent front positions. This will make Fig. S4 obsolete, so we propose to remove that figure.

More generally, I would say that the observations presented do not represent clear cut evidence of a transition from one calving style to the next. For example, given the stochastic nature of calving I would be a little cautious in ascribing too much meaning to a few individual events that happen pre/post initiation of melt, even though they may be substantial. What is more convincing for me is the combination of the previous studies (Jouvet et al., 2017; van Dongen et al., 2020a,b) identifying that large scale calving events occurred due to hydrofracture/melt undercut, the arguably circumstantial evidence presented here, and evidence for recent ungrounding combined with GPS data showing evidence of calving by buoyant flexure. I think the discussion would benefit if these lines of evidence were linked more strongly.

We agree that the observations do not represent a clear transition, which is why we deliberately avoided the use of the word 'transition' and rather expressed that the observations suggest that buoyant calving has occurred more frequently. We do not argue that *all* recent calving events were a result of buoyant forces, but that buoyant calving has been playing a more important role since 2018, responsible for a larger part of the mass loss by calving.
In order to more clearly link the evidence from the previous studies to the findings in this paper, we will move the first paragraph of Sect 5.2 (L299-301) to Sect. 5.4, and add more information about the mentioned previous studies, such that Sect. 5.4 starts with:

*"Numerical modelling showed that an undercut is necessary in order to reproduce the initiation of the crevasse which lead to calving in 2017 (van Dongen et al., 2020b). Furthermore, high-resolution terrestrial radar interferometry data revealed that crevasse opening prior to calving was fastest at low tide, which suggests a non buoyant calving style (van Dongen et al., 2020a). Additional modelling work identified the crevasse water level, and thus hydro-fracturing, as a key driver of opening rates (van Dongen et al., 2020a). Unlike the deep and most likely water-filled surface crevasses which formed prior to calving in 2015 and 2017 (Fig. 2), no precursor crevasse was visible in the field prior to the major calving events in July 2019. This suggests that the calving events in July 2019 were not triggered by extensional stress at the glacier surface, but resulted from basal crevasse propagation due to buoyant flexure instead."*

New references
Gorelick, N., Hancher, M., Dixon, M., Ilyushchenko, S., Thau, D., and Moore, R.: Google Earth Engine: Planetary-scale geospatial analysis for everyone, Remote sensing of Environment, 202, 18–27, https://doi.org/10.1016/j.rse.2017.06.031, 2017.
Strozzi, T., Luckman, A., Murray, T., Wegmuller, U., and Werner, C. L.: Glacier motion estimation using SAR offset-tracking procedures, IEEE Transactions on Geoscience and Remote Sensing, 40, 2384–2391, https://doi.org/10.1109/TGRS.2002.805079, 2002.

---

## Author Response (AR2)

Dear Olaf,

Thank you very much for accepting our manuscript for publication in The Cryosphere. The word "Projections" only appeared falsely in the marked-up version of the manuscript, in which the changes in the "Data availability" section were not automatically detected by the software latexdiff, and something went wrong when manually highlighting the changes in this section. Therefore, the uploaded manuscript is equal to the version uploaded on 14 Dec.

Thank you for the rapid handling of our manuscript, we appreciate this very much.

Best regards, on behalf of all co-authors,
Eef